# Dendritic Cell-Based Cancer Vaccines: The Impact of Modulating Innate Lymphoid Cells on Anti-Tumor Efficacy

**DOI:** 10.3390/cells14110812

**Published:** 2025-05-30

**Authors:** Yeganeh Mehrani, Solmaz Morovati, Fatemeh Keivan, Soroush Sarmadi, Sina Shojaei, Diba Forouzanpour, Byram W. Bridle, Khalil Karimi

**Affiliations:** 1Department of Pathobiology, Ontario Veterinary College, University of Guelph, Guelph, ON N1G 2W1, Canada; ymehrani@uoguelph.ca; 2Division of Biotechnology, Department of Pathobiology, School of Veterinary Medicine, Shiraz University, Shiraz 71557-13876, Iran; s.morovati@shirazu.ac.ir; 3Department of Microbiology and Immunology, School of Veterinary Medicine, University of Tehran, Tehran 14179-35840, Iran; fatemeh.keivan@ut.ac.ir (F.K.); soroush.sarmadi@ut.ac.ir (S.S.); sina.shojaei@ut.ac.ir (S.S.); d.forouzanpour@ut.ac.ir (D.F.)

**Keywords:** dendritic cell (DC) vaccines, innate lymphoid cells (ILCs), tumor microenvironment (TME)

## Abstract

Dendritic cell (DC) vaccines stimulate the immune system to target cancer antigens, representing a promising option for immunotherapy. However, clinical trials have demonstrated limited effectiveness, emphasizing the need for enhanced immune responses. Improving the production of DC vaccines, assessing their impact on immune components, and observing responses could improve the results of DC-based therapies. Innate lymphoid cells (ILCs) represent a heterogeneous population of innate immune components that generate cytokines and modulate the immune system, potentially enhancing immunotherapies. Recent research highlights the different functions of ILCs in cancer, demonstrating their dual capabilities to promote tumors and exhibit anti-tumor actions. DCs and ILCs actively communicate under physiological and pathological conditions, and the activation of ILCs by DCs or DC vaccines has been shown to influence ILC cytokine production and function. Gaining insights into the interaction between DC-activated ILCs and tumors is essential for creating exciting new therapeutic strategies. These strategies aim to boost anti-tumor immunity while reducing the support that tumors receive. This review examines the effect of DC vaccination on host ILCs, illustrating the complex relationship between DC-based vaccines and ILCs. Furthermore, it explores some exciting strategies to enhance DC vaccines, aiming to boost anti-tumor immune responses by fostering better engagement with ILCs.

## 1. Introduction

Cancer immunotherapy provides targeted and personalized treatment options that empower the immune system to fight cancer [1]. Accordingly, dendritic cell (DC)-based vaccines initiate and regulate innate and adaptive immune responses to combat cancers [2]. DCs support innate immunity by interacting with various innate lymphocytes, such as natural killer (NK) and T cell receptor (TCR)-γδ cells [3], fighting against cancers [4]. Furthermore, DCs serve as key antigen-presenting cells (APCs) and are essential in activating cytotoxic T lymphocyte cells (CTLs) and T-helper cells, which contribute to tumor elimination [5]. Even though cancer vaccines developed in DC show promising potential, their clinical effectiveness remains beyond expectations. Therefore, strategies are necessary to enhance their immunogenicity and overcome tumor-induced immune suppression [6].

Recent studies emphasize the critical role of innate lymphoid cells (ILCs) in shaping immune responses within the tumor microenvironment (TME) [7]. ILCs are a diverse family of innate immune cells contributing to tumor immunity through cytokine production and interactions with other immune cells [8]. Among the ILCs, ILC2s and ILC3s have been shown to contribute to both pro-tumor and anti-tumor mechanisms, raising questions about their potential modulation during DC-based vaccination [9]. The interaction between DCs and ILCs may influence the strength and quality of the anti-tumor immune response, making these cells promising targets for therapeutic strategies [10,11].

This review explores the complex relationship between DC-based cancer vaccines and host ILCs, emphasizing how modulating various ILC subsets can improve vaccine effectiveness. Exploring the dynamic interaction between DCs and ILCs could lead to innovative combined strategies that enhance the therapeutic efficacy of DC-based vaccines, ultimately improving clinical outcomes for cancer patients.

## 2. Characterization of Dendritic Cell Vaccines: Evidence of Antitumor Efficacy

### 2.1. Dendritic Cell Subsets: Distribution and Functional Roles

DCs are antigen-presenting cells essential for initiating and regulating adaptive immune responses. Derived from hematopoietic stem cells, DCs differentiate into various subtypes across tissues and mature in lymphoid organs. Their ability to activate naïve T cells and modulate immune responses makes them key targets for cancer immunotherapies [12].

DCs include conventional DCs (cDCs), plasmacytoid DCs (pDCs), monocyte-derived DCs (Mo-DCs), and Langerhans cells (LCs) (Table 1) [13,14].

cDC1 cells specialize in cross-presenting exogenous antigens to CD8^+^ CTLs through MHC class I molecules (MHCI), playing a crucial role in anti-tumor and antiviral immunity [15,16].

cDC2 cells secrete cytokines such as interleukin-10 (IL-10), IL-12, IL-23, and tumor necrosis factor-β (TNF-β), which are known to drive the differentiation of CD4^+^ helper T cells (Th) and play a crucial role in initiating and regulating humoral immunity, inflammatory responses, and immune tolerance [17].

pDCs secrete Interferon (IFN)-α and IFN-β, essential for several immune processes, including anti-tumor activity [18]. Moreover, IFNs enhance adaptive immunity by priming T cells, increasing their responsiveness to antigen presentation. Specifically, IFN-α and IFN-γ promote DC maturation, leading to the upregulation of co-stimulatory molecules CD80, CD86, and CD40, which are essential for optimal T cell activation and differentiation [19].

Mo-DCs, derived from circulating monocytes responding to inflammation and immune signals, primarily accumulating in inflamed tissues. They play a crucial role in antigen presentation and immune regulation as a key link between innate and adaptive immunity [20].

LCs are a specialized subset of DC that predominantly reside in the skin’s epidermis and certain mucosal surfaces, serving as the first line of defense in skin and mucous membranes [21]. Beyond their role in stimulating the immune system, LCs also help maintain immune tolerance. Regulatory T cells (Tregs) help suppress excessive immune responses and prevent autoimmune diseases [22].

The aforementioned unique properties of DC subsets position them as a valuable platform for cancer vaccines, especially when paired with immunomodulatory strategies to enhance their therapeutic efficacy. Furthermore, the examined DC vaccines demonstrated their potential in cancer immunotherapy but still face certain challenges that must be addressed. One key challenge is to enhance the reproducibility of DC vaccine effectiveness, which relies significantly on optimal DC maturation and their interactions within the TME. Future research will focus on improving DC differentiation, identifying the best DC subsets for specific cancers, and exploring combination regimens with other immunomodulatory strategies to overcome current limitations. Understanding how to modulate these immune cells more effectively could significantly improve the therapeutic effectiveness of DC vaccines.

**Table 1 cells-14-00812-t001:** Overview of Dendritic Cell Types, Main Locations, and Key Functions.

Type of DC	Main Location	Key Functions
cDC1	Peripheral Tissue and Lymph [23]	Activation of CD8+ T cells [16]
cDC2	Peripheral Tissue and Lymph [23]	Activation of CD4+ T cells [17]
pDCs	Lymphoid Tissue and Blood [24]	Production of IFNα/β [14,25]
Mo-DCs	Inflamed Tissue [20]	Response to infection and inflammation [20]
LDC	Skin, Mucous Membranes [21]	Cutaneous and mucosal immunity [21]

### 2.2. Generation of DC Vaccines

Regardless of the types of DC-based vaccines, the production of DC vaccines involves several key steps, as illustrated in Figure 1. Initially, DC precursors are extracted from peripheral blood and cultured to yield autologous iDCs. Subsequently, these iDCs undergo maturation in vitro by introducing a mix of cytokines, toll-like receptor (TLR) agonists, and other stimulatory substances, such as granulocyte-macrophage colony-stimulating factor (GM-CSF), IL-4, IL-6, TNF-α, prostaglandin E2 (PGE2), poly (I:C), LPS, or CD40 ligand [26,27,28,29]. After maturation, tumor-associated antigens (TAAs) are introduced to the DCs through co-culturing with peptides, proteins, tumor lysates, or whole tumor cells. Finally, the matured antigen-loaded DCs are harvested and administered to the patient as a therapy vaccine (Figure 1).

### 2.3. Dendritic Cell Vaccines

DC-based vaccines utilize the antigen-presenting capacity of DCs to prime tumor-specific T cells and enhance adaptive immunity. Preclinical and clinical studies have demonstrated their potential in cancer immunotherapy [30]. These findings are discussed as follows.

#### 2.3.1. Ex Vivo DC-Based Vaccines

Ex vivo DC-based vaccines represent a form of personalized immunotherapy that takes advantage of DCs’ ability to activate adaptive immune responses. These vaccines are developed by isolating and ex vivo modifying DCs, then reintroducing them after they have been primed with tumor or pathogen antigens. Ex vivo DC-based vaccines have been extensively researched for their potential in cancer immunotherapy and treating infectious diseases [31]. These vaccines enhance antigen presentation, which activates CTLs and CD4^+^ T cells, leading to a potent anti-tumor response [31]. Overall, these vaccines hold significant potential in treating tumors [32].

#### 2.3.2. In Vivo DC-Targeting Vaccines

In vivo DC-targeting vaccines represent an advanced immunotherapy strategy that directly activates DCs within the body, resulting in a potent and specific immune response. These vaccines utilize antigen delivery systems, molecular adjuvants, and targeted receptor binding to enhance immune activation against infectious diseases and cancer [33].

#### 2.3.3. Plasmid DNA-Based DC Vaccines

Plasmid DNA-based DC vaccines utilize genetically engineered plasmid DNA to encode specific antigens, facilitating direct antigen expression in DCs and inducing a potent immune response [34]. Upon internalization by DCs, plasmid DNA is transcribed and translated into antigenic proteins, which are subsequently processed and presented on MHC molecules [35].

#### 2.3.4. Virus-like Particle (VLP)-Based DC Vaccines

VLP-based DC vaccines utilize the immunogenic properties of VLPs to enhance antigen presentation and immune activation. These self-assembling, non-replicative structures mimic viruses but lack genetic material, ensuring safety and strong immunogenicity [36]. Upon activation, DCs migrate to lymph nodes and prime naïve T cells, initiating CTL-mediated tumor or pathogen elimination and CD4^+^ Th cell-driven antibody production and memory formation [37].

#### 2.3.5. RNA-Based DC Vaccines

RNA-based DC vaccines rely on messenger RNA (mRNA) or other RNA molecules to enable DCs to express specific antigens, stimulating a strong adaptive immune response. This approach capitalizes on the natural role of DCs as APCs and has been investigated in clinical trials for cancer immunotherapy, viral diseases, and other pathological conditions [38]. This transfection can be achieved using methods such as electroporation, liposome-based delivery, or nanoparticle-mediated transfer [39]. The RNA-loaded DCs are stimulated with cytokines (including GM-CSF, IL-4, and TNF-α) to enhance their antigen-presenting capacity before being reintroduced into the patient [40]. RNA-based DC vaccines represent a powerful and flexible approach to immunotherapy, providing a highly personalized, safe, and effective method for combating cancer and infectious diseases [38].

#### 2.3.6. Peptide-Based DC Vaccines

Peptide-based DC vaccines are prepared by activating DCs with short antigenic peptides. Once activated, these DCs induce an adaptive immune response [41].

#### 2.3.7. Autologous DC-Based Vaccines

Autologous DC-based vaccines represent a personalized immunotherapeutic strategy designed to activate a patient’s immune system against cancer and infectious diseases [42]. These DCs migrate to lymphoid organs, prime naive T cells, initiate antigen-specific immunity, and ensure immediate and long-term immune responses [43].

#### 2.3.8. Allogeneic DC Vaccines

Allogeneic DC vaccines, derived from healthy donors, provide a standardized approach to cancer immunotherapy and infectious disease treatment by triggering an adaptive immune response that targets tumor or infected cells, thereby strengthening immune defense [44,45]. One of the critical factors to consider in allogeneic DC vaccine strategies is the immunological compatibility between the donor and the recipient, particularly concerning human leukocyte antigen (HLA) haplotypes. Unlike autologous DCs, which match the patient’s immune profile, allogeneic DCs present antigens via donor-derived HLA molecules [46]. If these HLA molecules do not adequately match those of the recipient, it can lead to inefficient antigen presentation, resulting in suboptimal T cell priming, or even trigger alloimmune responses, such as the production of anti-donor antibodies or cytotoxic T cell responses against the vaccine itself [47]. These immunological mismatches can compromise both the safety and therapeutic effectiveness of the vaccine. Consequently, consideration of full or partial HLA compatibility is essential when employing allogeneic DC vaccines to enhance immunogenicity while minimizing potential adverse effects [48].

### 2.4. Strategic Selection of DCs to Enhance Immunotherapeutic Outcomes

Among the various DC subtypes, Mo-DCs are the most widely utilized due to the high abundance of monocytes in peripheral blood. However, Mo-DCs exhibit lower CTL activation potency, e.g., in melanoma immunotherapy, than LCs [49,50]. Additionally, cDCs demonstrate superior capacity in inducing systemic immune responses and enhancing the efficacy of immune checkpoint blockade therapy [51].

Regardless of technological advancements in flow cytometry and immune bead sorting that enable the selective enrichment of specific DC subtypes to optimize targeted CTL activation, the ideal DC subtype for vaccine development remains unclear [52]. Due to the limited availability of cDCs for extensive vaccine production, most clinical trials investigating DC-based vaccines rely on Mo-DCs [53]. However, Mo-DC cultures are not often a heterogeneous population [54,55]. Despite these limitations, Mo-DCs remain the primary cell type in clinical trials for DC-based vaccines targeting ovarian cancer, highlighting the need for further research to optimize DC subtype selection and vaccine efficacy [56].

Although ex vivo DC vaccines are effective, they are limited by technical challenges, including cell isolation and modification. Conversely, in vivo DC-targeting vaccines offer advantages in broader clinical applications but require further optimization, particularly regarding antigen delivery efficiency.

Future research may address the comparative effectiveness of autologous versus allogeneic DC vaccines. Autologous DC vaccines offer personalized treatment but are associated with substantial costs and operational challenges. In contrast, allogeneic DC vaccines provide standardized production but present potential risks due to immunological incompatibility. Additionally, traditional Mo-DCs exhibit limited capacity to activate CTLs, necessitating the exploration of alternative DC subsets, including cDCs and LCs, for enhanced therapeutic outcomes. In recent years, numerous studies have investigated dendritic cell vaccination for treating hematological malignancies, such as acute myeloid leukemia, myelodysplastic syndromes, and lymphoma, as well as non-leukemic malignancies like prostate cancer, lung cancer, glioma, and gastric cancers [57,58].

Despite current limitations, DC-based vaccines remain a valuable approach for enhancing anti-tumor immune responses. Advancements in next-generation DC vaccine platforms, combined with complementary immunotherapeutic strategies, such as immune checkpoint inhibitors (anti- programmed death-1 (PD-1)/PD-L1), cytokine therapies (IL-2, GM-CSF), and adoptive T cell transfer, hold the potential to improve clinical outcomes for cancer patients significantly.

## 3. Limitations and Challenges in DC-Based Immunotherapy

DCs vaccines have been used in clinical trials as cellular therapeutics for cancer immunotherapy since the mid-1990s. DC-based immunotherapeutic approaches have been shown to have safety profiles and the ability to induce anti-tumor immune responses, including in patients with late-stage malignancies. Nevertheless, the clinical outcomes have been predominantly unsatisfactory, with conventional measurable tumor response rates typically remaining below 15% [50]. Nonetheless, the pursuit of strategies for advancing and evaluating DC-based vaccines persists despite the challenges associated with limited effectiveness [59]. Consequently, current efforts primarily focus on new strategies designed to significantly enhance the efficacy and therapeutic applicability of these DC-based therapies. A meta-analysis of DC-based vaccines for prostate cancer and renal cell carcinoma systematically reported that these therapies induced immune responses in 77% and 61% of patients with prostate cancer and renal cell carcinoma, respectively [60]. In addition to adaptive immunity enhancement, it was demonstrated that DCs enhance the anti-tumor function of natural killer cells (NK cells) by increasing their cytolytic activity and the induction of IFN-γ release [61]. Several studies have emphasized the pivotal role of DC-NK cell cross-talks in effective tumor elimination [30]. Effective anti-tumor immunity mediated by DC vaccines relies partially on NK cell activity. This was demonstrated in murine melanoma and metastatic lung tumor models, where the tumor elimination observed after DC vaccination was suppressed entirely in the animals that were depleted of NK cells [62,63]. Although few DC-based cancer vaccination trials have incorporated NK cell recruitment, data indicate that nearly 50% of patients experienced increased NK cell frequency or enhanced NK cell activation [30]. The anti-tumor responses enhanced by DC vaccination have been shown to depend critically on DC-NK cell cross-talks, which play a key role in tumor elimination [64]. These findings highlight the essential function of DC-NK cell cross-talk in the effectiveness of DC-based cancer vaccines. Future strategies may include incorporating NK cell-activating adjuvants, modulating cytokine signaling pathways like IL-15, and using engineered DCs with improved capacity for NK cell engagement to amplify anti-tumor immunity.

Clinical studies have demonstrated that DCs loaded with apoptotic tumor cells induced more potent immune responses than DCs loaded with tumor lysates or RNA-pulsed DCs [65,66]. Although tumor lysates or apoptotic bodies serve as valuable sources of antigenic substances, such advantages are limited by the availability of tumor cells. Moreover, monitoring the immune responses is more complex, making the correlation of the interventions against clinical outcomes challenging. The effectiveness of DC-based immunotherapy relies on crucial DC characteristics, especially maturation status and cytokine profile [67]. Clinical studies consistently show that mature DCs exhibit enhanced therapeutic effectiveness compared to iDCs [60,68]. This is mainly because iDCs have poor migratory capacity from injection sites to lymph nodes and can suppress CTL function by inducing antigen-specific IL-10-producing Tregs [68,69]. In addition to maturation, the cytokine and chemokine profiles of DCs, particularly IL-12 and CCR7 expression, are critical for vaccine efficacy. IL-12 is essential for CD8+ T cell proliferation and the differentiation of CTL [70]. However, DCs do not always produce sufficient levels of IL-12, which may limit their immunostimulatory potential [71]. Clinically, higher levels of IL-12 derived from DCs in these vaccines for glioma and melanoma patients have been associated with better clinical outcomes [72,73].

CCR7, a key chemokine receptor, transmits DC from peripheral tissues to draining lymph nodes, ensuring effective antigen presentation. This migration is crucial for optimal interaction with T cells, which is necessary for a potent immune response [74,75]. Notably, DCs with high CCR7 expression induce stronger antigen-specific immunity in vivo, allowing for a lower vaccine dose while maintaining effectiveness; however, not all DCs in vaccines express adequate levels of CCR7, which may limit their migratory efficiency and overall immunogenicity [75,76]. Moreover, the therapeutic potency of DCs relies heavily on their maturation status and cytokine profile, particularly IL-12 and CCR7 expression, which are key for effective CTL activation and lymph node homing. Therefore, future strategies should focus on standardized maturation protocols and adopt molecular or genetic enhancements to upregulate IL-12 and CCR7 reliably. These advancements are essential to improve DC-based immunotherapies’ consistency and clinical impact.

One of the key challenges limiting the efficacy of DC-based vaccines is the lack of a universally optimal maturation stimulus [77]. The ideal maturation stimulus for DCs is still under investigation. Various maturation strategies have been evaluated, employing individual factors or combinations of pro-inflammatory cytokines, CD40 ligand (CD40L), and TLR agonists. However, no single method has consistently demonstrated greater therapeutic effectiveness in clinical settings [33]. The most commonly used maturation cocktail in clinical trials includes TNF-α, IL-1β, IL-6, and PGE2. However, PGE2 exhibits a paradoxical effect, as it enhances the migratory capacity of DCs by upregulating CCR7 expression while simultaneously reducing IL-12 secretion [78,79]. Another common DC maturation cocktail includes TNF- α, IL-1b, IFN-α, and IFN-γ. The resulting DCs exhibit enhanced IL-12 secretion and have been shown to induce more potent CTL responses compared to conventionally matured DCs [72,80,81,82]. Despite these advancements, heterogeneity in maturation strategies remains a significant challenge for improving the clinical efficacy of DC vaccines [83].

DC vaccines are typically administered along with adjuvants to improve their effectiveness. The most common adjuvants administered are GM-CSF, IL-2, IFN-α, and TLR agonists [67]. GM-CSF plays a pivotal role in stimulating the proliferation of DC precursors and also has a potent chemotactic action, which promotes the recruitment and maturation of DCs. IFN-α enhances the antigen cross-presentation activity of DCs, resulting in more effective CTL responses [84]. Although preclinical studies have demonstrated that IL-2 can improve the efficacy of DC vaccines [85], subsequent clinical trials have failed to demonstrate an increase in anti-tumor immune responses through DC administration combined with IL-2 [67,86]. IL-2 promotes the expansion of Tregs and myeloid-derived suppressor cells (MDSCs), which may impair the efficacy of DC-based immunotherapies by creating an immunosuppressive environment. Moreover, TLR agonists are increasingly employed as adjuvants in DC vaccines, with their incorporation into maturation protocols steadily rising to counteract these immunosuppressive effects [67]. Future investigations should prioritize developing defined maturation protocols that enhance DC migratory capacity, primarily through CCR7 upregulation, without compromising the secretion of key pro-inflammatory cytokines such as IL-12, which are critical for effective Th1 polarization and cytotoxic T lymphocyte activation. Concurrently, optimizing the selection and timing of adjuvant administration is essential to mitigate the induction of immunosuppressive populations, including Tregs and MDSCs. Integrating TLR agonists and type I IFNs into maturation regimens represents a strategy to improve DC immunogenicity while attenuating tolerogenic signaling pathways.

A significant limitation of DC vaccines is the inefficiency of current delivery methods in ensuring efficient migration to lymphoid tissues, where effective T-cell priming occurs [87]. The delivery method, administration frequency, and dose of DCs are critical determinants of DC-based vaccine efficacy, yet no single method has demonstrated consistent superiority [88]. While intravenous (IV) injection targets the lungs, liver, and spleen, with limited lymph node involvement, and is more effective for humoral responses in prostate cancer studies [89]. In contrast, intradermal (ID) and subcutaneous (SC) injections keep most DCs at the injection site, but 1–5% migrate to lymph nodes to activate T-cells [87,89].

The search for an optimal delivery strategy has led to an increasing trend toward combining multiple administration routes, such as ID, IP, SC, IV, intra-lymphatic, and intranodal injection, to enhance vaccine efficacy [67,90,91,92,93]. However, these combination strategies continue to face clinical challenges, and the effectiveness of various vaccine administration methods has varied across different tumor models. An ideal approach should achieve a balance between immunogenicity and safety. Current research should focus on systematically refining DC maturation and adjuvant strategies to improve lymphoid tissue trafficking and enhance functional immunogenicity, thereby increasing clinical efficacy in cancer immunotherapy.

Immune-suppressive mechanisms in cancer patients significantly hinder the efficacy of DC-based immunotherapy. These mechanisms inhibit the functional competence of CTLs and NK cells. In particular, the immune checkpoint receptors cytotoxic T-lymphocyte–associated antigen 4 (CTLA-4) and PD-1 are key factors in negative regulation of CTL function [94].

The development of monoclonal antibodies targeting these inhibitory pathways has introduced opportunities for combination strategies with DC immunotherapy. Preliminary clinical data from patients with advanced melanoma indicate that combining DC therapy with CTLA-4-targeting monoclonal antibodies enhances therapeutic efficacy compared to monotherapy [95,96]. However, the potential for immune-related toxicity necessitates a cautious approach when integrating these agents into treatment protocols [50].

To improve the effectiveness of DC vaccines, clinical trials must systematically compare DCs matured using various stimulation protocols to establish a standardized approach, given the current variability in maturation strategies across trials. Their efficacy may be further enhanced through combination with other therapeutic strategies, a method aimed at maximizing their immunostimulatory potential. Despite its current limitations, DC vaccine technology demonstrates significant potential for personalized medicine and clinical safety, particularly when integrated with adjunctive therapies [59].

DC-based vaccines often show limited success in cancer treatment due to the complexity and variability of the TME. The TME includes many types of cells and signaling molecules that change from one tumor region to another and from one patient to the next [97,98]. In solid tumors, high numbers of suppressive immune cells, such as Tregs, MDSCs, and M2 macrophages, can block the activity of DCs and the T cells they aim to activate [99,100]. Other TME factors like hypoxia and acidic conditions also weaken the ability of DCs to mature and present antigens [101]. Furthermore, the uneven presence of checkpoint molecules like PD-L1 across the tumor can result in inconsistent T cell activation [102]. These challenges suggest that using DC vaccines alone may not be enough. Combining them with other treatments, such as checkpoint inhibitors or drugs that target suppressive immune cells, may improve outcomes [14,103].

## 4. ILCs in Oncology

### 4.1. Overview of ILCs—Development and General Functions

ILCs have recently been categorized into three distinct subsets, each with specific roles in immune regulation across various tissues and conditions [104]. The ILCs family includes ILC1, ILC2, ILC3, NK cells, and lymphoid tissue inducer (LTi) cells, each with distinct roles in the immune system [105]. ILC1s and NK cells together form group-1 ILCs. ILC3s are subdivided into three populations based on their expression of CCR6 and natural cytotoxicity receptor (NCR, i.e., NKp46 in mice and NKp44 in humans). CCR6^+^ cells are designated LTis and produce lymphotoxins (LTs). CCR6^-^ cells are divided into NCR^+^ ILC3s and NCR^-^ILC3s [106].

ILC1s express the transcription factor T-bet and are essential for producing IFN-γ and TNF-α, which help combat intracellular pathogens, similar to Th1 cells. They include NK cells, which also exhibit cytotoxic functions against virus-infected cells and tumors, and non-NK ILC1s, which primarily mediate inflammation and immune responses against bacterial and parasitic infections [107].

Group 2 ILCs (ILC2s) characterized by high expression of GATA3, ILC2s produce cytokines such as IL-4, IL-5, IL-9, and IL-13 in response to helminthic infections and allergens, similar to Th2 cells that produce similar cytokines. These cells play crucial roles in mucosal immunity, tissue repair, and allergic responses [108].

Group 3 ILCs (ILC3s) express RORγt and serve as key producers of IL-17 and IL-22, resembling Th17 and Th22 cells in their cytokine production. These cytokines play a crucial role in gut immunity and inflammation. ILC3s are essential for lymphoid tissue formation during immune development [109,110,111,112,113]. Among ILC3s, LTi cells play a pivotal role in the early development of lymphoid structures, including lymph nodes and Peyer’s patches [114]. These cells contribute to the structural organization of lymphoid tissues and influence adaptive immune responses [115].

Evidence indicates that ILCs regulate immune responses in the TME, playing dual roles in either combating or facilitating tumor progression [116] (Figure 2 and Figure 3). Tissue-resident ILCs are strategically positioned within tumors, enabling them to respond rapidly and modulate immune activity against malignancies. Given their ability to either promote or inhibit tumor progression, understanding the functional diversity of ILC subsets is crucial for optimizing their therapeutic potential.

### 4.2. Innate Lymphoid Cells and Tumor Immunity: Roles and Mechanisms of Action

ILCs are highly responsive to signals within the TME, including cytokines, chemokines, and metabolic changes. These signals can profoundly affect the differentiation, survival, and function of ILCs [116].

Like T cells, ILCs express various immune checkpoint molecules such as PD-1 and CTLA-4. The TME can manipulate the expression of these molecules to suppress ILC activity and prevent their anti-tumor effects [117]. Modulating ILC activity through cytokine therapy or targeting immune checkpoints (e.g., PD-1, Lymphocyte activation gene 3 protein (LAG3), T cell immunoglobulin, and mucin domain-containing protein 3 (TIM-3)) offers potential pathways to shift the balance between pro-tumor and anti-tumor activities of ILCs in the TME [117] (Figure 2 and Figure 3).

Furthermore, ILCs demonstrate notable functional plasticity; ILC1s can convert into ILC3-like cells in the presence of certain cytokines like IL-23 and IL-1β, altering their cytokine profile and function within the TME [108]. Transforming growth factor-β (TGF-β) signaling has been shown to promote ILC plasticity, shifting the balance toward a pro-tumor state. ILC1s and NK cells express inhibitory receptors, such as natural killer group 2 member A (NKG2A), Killer cell lectin-like receptor G1 (KLRG1), and other inhibitory receptors, which contribute to tumor advancement in an immunosuppressive TME [7,118]. This adaptive capability of ILCs suggests potential therapeutic strategies, such as modulating the activity of TGF-β to regulate the dynamics between ILC2s and ILC3s, potentially enhancing oncological treatment outcomes.

#### 4.2.1. ILC1: The Cytotoxic Regulators of Tumor Immunity

ILC1s are recognized for producing IFN-γ and TNF-α, sharing functional parallels with NK cells. These cells can play tumor-suppressing and tumor-promoting roles depending on the tumor type and signals from the tumor environment. Helper ILC1s are similar to NK cells. These cells share many receptors, making it difficult to distinguish between them. ILC1s and NK cells express T-bet, NKp46 receptor, and produce IFN-γ [119]. However, NK cells have more potent cytotoxic effects. They actively detect and destroy virus-infected tumor cells. Moreover, these cells possess potent cytotoxic activity, utilizing molecules such as perforin and granzyme to kill target cells. Unlike NK cells, ILC1s have limited ability to kill target cells. Instead, they contribute more to immune regulation and inflammation through cytokine production, particularly IFN-γ [120,121].

ILC1s can contribute to tumor support within the TME. Typically involved in immune defense, these cells can be influenced by high levels of TGF-β, which are often found in tumors. This influence prompts NK cells to transition into ILC1-like cells, adopting less effective characteristics at combating tumor growth [122]. These intermediate ILC1s are characterized by the expression of CD49a and CD49b, distinguishing them from the classic NK cells. They exhibit higher expression levels of immunosuppressive inhibitory receptors, such as CTLA-4, LAG3, and CD96, which suppress immune responses. Consequently, this transformation reduces their cytotoxic capabilities, which are crucial for attacking and eliminating cancer cells, thereby inadvertently facilitating the progression and spread of tumors, such as fibrosarcoma [123,124]. They also produce GM-CSF, which promotes the growth of myeloid cells and contributes to the inflammatory response. Additionally, these cells produced low amounts of IFN-γ and CCL5 [122]. All of these elements contribute to the creation of a pro-tumor environment. Furthermore, TNFα, which is produced by these ILC1s, is associated with immune escape [122]. Additionally, TGF-β signaling has been shown to have different effects in various environments, for instance, enhancing immune responses in healthy salivary glands but potentially contributing to tumor progression in salivary glands with cancer [125,126]. Researchers have identified a specific group of ILC1s that express NKp46^+^ and NK1.1^+^ in patients with non-small cell lung cancer (NSCLC). These cells exhibited lower levels of Eomesodermin (Eomes), diminished cytotoxicity, and reduced IFN-γ production. This functional decline has been linked to IL-12 and IL-18 signaling, which promotes the development of less responsive ILC1s with attenuated effector function capacity [127]. This adaptation highlights the complex role of ILC1s in cancer, where they may support the tumor by reducing the immune system’s ability to target malignant cells aggressively.

On the other hand, ILC1s primarily produce IFN-γ and TNF-α. IFN-γ acts directly on tumor cells to increase the expression of MHC I molecules, enhancing their recognition by CD8^+^ T cells [128]. TNF-α, produced by ILC1s, plays a significant role in tumor suppression. It directly destroys tumor cells and promotes cell death and pyroptosis. This activity indirectly recruits and stimulates anti-tumor macrophages and DCs to enhance anti-tumor responses. Additionally, TNF-α is crucial for the induction and activation of CTLs, among the most effective cells at killing tumors [129].

#### 4.2.2. ILC2: Orchestrators of Type 2 Immunity in the Tumor Microenvironment

ILC2s play a complex role in tumor immunity, promoting or suppressing cancers. For instance, the generation of IL-13 by ILC2s could contribute to negative outcomes in some cancers, while in others, releasing IL-5 enhances anti-tumor responses [130]. ILC2s secrete type 2 cytokines, including IL-5, IL-13, IL-9, and amphiregulin, which can promote tumor growth. The cytokines from ILC2s facilitate the recruitment and activation of Tregs and MDSCs, both of which suppress anti-tumor immunity in various cancers by activating tumor-promoting pathways, such as those in breast, prostate, and gastric cancers [107,131,132,133]. ILC2s promote tumor progression by secreting IL-4 and IL-13, which activate the IL-4Rα–STAT6 signaling pathway. This pathway enhances the recruitment of Foxp3^+^ Tregs and the polarization of CD11b^+^ cells into M2 macrophages, contributing to an immunosuppressive tumor microenvironment. In lung and renal cancers, elevated IL-4/IL-13 levels are linked to increased tumor growth, metastatic potential, and poor clinical outcomes [134,135,136].

Chronic overactivation of ILC2s, particularly under the influence of cytokines such as IL-33 and IL-25, exacerbates this pathology by driving the sustained production of amphiregulin and IL-13 [137]. Notably, prolonged fibrotic responses may evolve into malignancy, as seen in hepatic fibrosis, progressing to cirrhosis and, ultimately, hepatocellular carcinoma [138,139,140]. These findings highlight the potential of ILC2s to contribute to tumorigenesis through fibrosis-driven mechanisms associated with chronic inflammatory conditions or persistent tissue repair processes.

ILC2s can also promote the accumulation of Tregs through the interaction of inducible T-cell co-stimulator ligand (ICOSL) on ILC2s and inducible T-cell co-stimulator (ICOS) on Tregs. This interaction helps increase the number of Tregs in the TME, further promoting tumor growth and immune suppression [141,142].

In patients who are in remission from acute promyelocytic leukemia (APL), there is a decrease in ILC2s, IL-13, prostaglandin D2 (PGD2), and MDSCs [133]. The ILC2/M-MDSC interaction plays a significant role in tumor suppression by the immune system. In mice with tumors, PGD2 (produced by tumor cells) interacts with CRTH2 (a receptor expressed by ILC2s), and B7H6 interacts with NKp30, another receptor expressed by ILC2s. This activates ILC2s, causing them to produce higher levels of IL-13. IL-13 binds to its receptor (IL-13Rα1) on M-MDSCs, activating these suppressor cells. Activated M-MDSCs create an immunosuppressive environment that helps tumor growth and evades immune attacks [143]. Researchers have also blocked PGD2, IL-13, and NKp30, reducing ILC2s and M-MDSCs. The improved survival rates in these animals suggest that this pathway could be a target for cancer therapy [133].

IL-33 is a cytokine found in both mouse and human colorectal and breast cancers, which can activate ILC2s [144,145]. In mouse models of breast cancer (4T1 model), IL-33 increased the number of ILC2s, M-MDSCs, and Tregs. These cells create an immunosuppressive environment, helping tumor metastasis while promoting the formation of new blood vessels to feed the tumor [146].

ILC2s, while primarily known for their role in allergic inflammation, have shown beneficial effects in certain cancer models by contributing to tumor suppression. Specifically, recent studies demonstrate that KLRG1^+^ ILC2s can acquire anti-tumor functions through IL-10 production and IL-33–mediated activation. ILC2s, despite being primarily known for their role in allergic inflammation, have been shown to have beneficial effects in certain cancer models by contributing to tumor suppression. Specifically, recent studies have demonstrated that KLRG1^+^ ILC2s can acquire anti-tumor functions through IL-10 production and IL-33–mediated activation [147]. In lymphoma models, IL-33 enhances ILC2 accumulation and triggers the production of CXCR2 ligands by these cells. Concurrently, TME, characterized by hypoxia and oxidative stress, induces the expression of CXCR2 on tumor cells. The interaction between ILC2-derived ligands and CXCR2 on tumor cells leads to apoptosis [148].

ILC2s release GM-CSF, attracting eosinophils and potentially exerting anti-tumor effects by promoting their activation and recruitment through IL-5, thereby highlighting their influence within the TME [117]. Moreover, in a B16 melanoma mouse model, ILC2-derived IL-5 limited lung tumor establishment. IL-33 and IL-25 activation induced ILC2s to secrete IL-5, which recruited eosinophils, thereby promoting prolonged anti-tumor immune activity [149]. In patients with NSCLC, ILC2 levels are reduced in the lungs compared to healthy individuals. This observation suggests a potential protective role for ILC2s in lung cancer immunity [150,151]. Furthermore, ILC2s serve as the main source of IL-9 in a colorectal cancer (CRC) model. This cytokine not only stimulates CD8^+^ T cells but also inhibits tumor growth, suggesting a protective role for ILC2s [152]. Similarly, the IL-33–ILC2 pathway strengthens anti-tumor immunity in pancreatic cancer by mobilizing DCs and CD8^+^ T cells to attack cancer cells [150]. These findings suggest that under specific conditions, ILC2s can act as potent effectors of anti-tumor immunity. Realizing this therapeutic potential will require precise modulation of their activation and interactions within the TME.

#### 4.2.3. ILC3: The Dual Role in Tumor Progression and Immunity

ILC3s play key roles in mucosal immunity, and their cytokines can have dual effects on tumor progression [153].

The IL-23/IL-17 axis often promotes tumor growth by inducing an immunosuppressive environment that facilitates cancer progression [154]. IL-17A signaling within tumor cells triggers the secretion of CXCL5, leading to the subsequent recruitment of MDSCs [116]. ILC3s-derived IL-22 plays a complex role in tumor biology, exhibiting both pro-inflammatory and anti-inflammatory properties. This cytokine can promote tissue regeneration and support tumor survival and growth by enhancing cellular proliferation [155,156].

ILC3-driven CD4^+^ T cell activation occurs via the NF-κB signaling pathway, which regulates the immune response. However, TGF-β suppresses this function, suggesting that tumors may exploit TGF-β to evade immune detection by inhibiting ILC3-mediated T cell activation [157,158]. In the gut, ILC3s interact with T cells via major histocompatibility complex class II (MHC II) to regulate the microbiota and induce type 1 immunity. When ILC3s lose MHC II expression, mice develop colorectal cancer and become resistant to anti-PD-1 immunotherapy [159].

CCL21 plays a crucial role in shaping the TME by attracting NKp46⁻ ILC3 cells to the tumor. This recruitment stimulates stromal cells to produce CXCL13, facilitating interactions between ILC3 and stromal cells and possibly influencing tumor progression [160]. Also, NCR^+^ ILC3s contribute to angiogenesis in the TME by engaging the RANK–RANKL signaling pathways, primarily through interactions with endothelial and stromal cells within tertiary lymphoid structures [151].

ILC3s also serve as a significant source of pro-inflammatory cytokines, including IL-17 and GM-CSF. However, these cytokines can contribute to prolonged inflammation, which may subsequently lead to the initiation of tumors [161].

However, recent studies on ILC3s have highlighted their critical role in anti-tumor immunity. These cells facilitate an enhanced immune response against tumors by promoting the recruitment and activation of leukocytes at the tumor site. Specifically, ILC3 activation leads to an increased expression of MHC II molecules. This expression is crucial as it enhances the responsiveness of CD4^+^ and CD8^+^ T-cells, which are key players in controlling tumor growth [7].

Additionally, ILC3s contribute to effective antigen presentation by supporting or promoting the development of tertiary lymphoid structures (TLSs) within the tumor. These TLSs are crucial for orchestrating localized immune responses, as they facilitate the aggregation and functional coordination of immune cells, thereby enhancing their efficacy in targeting neoplastic cells [156]. The anti-tumor immune responses of ILC3s have been shown through multiple mechanisms, including direct cytotoxicity, antigen presentation, and the recruitment of other immune cells. Their activation is driven by cytokines such as IL-1β, IL-23, and IL-2, which stimulate ILC3s to secrete effector molecules including IFN-γ, GM-CSF, IL-8, IL-22, and TNF-α. These cytokines contribute to tumor control by promoting the recruitment of immune cells and amplifying inflammation. ILC3s can also recognize tumor cells and induce apoptosis in melanoma and hepatocellular carcinoma through TNF-related apoptosis-inducing ligand (TRAIL)-dependent pathways [162].

Louise Rethacker and colleagues identified a distinct ILC3 subset termed CD56^+^ILC3. This subset retains typical ILC3 characteristics while expressing cytotoxicity-associated genes, including those encoding perforin and granzymes, which are typically found in NK cells. CD56^+^ILC3 cells can adopt NK-like behavior upon cytokine exposure, enabling them to induce tumor cell apoptosis directly. Their dual functionality, which modulates immune responses and directly mediates tumor cell destruction, highlights their significant therapeutic potential in cancer treatment [163].

As insights into their plasticity and tissue-specific roles grow, ILCs are recognized as key regulators of tumor–immune interactions, underscoring the need for further research into their therapeutic potential in cancer immunology.

**Figure 2 cells-14-00812-f002:**
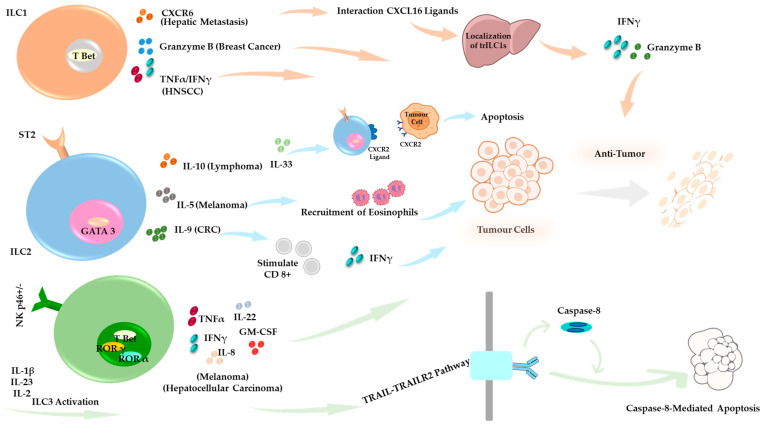
Anti-Tumor Roles of ILCs in the Tumor Microenvironment (TME). Under specific conditions, ILC subsets contribute to anti-tumor immune responses. ILC1s support tumor suppression through Granzyme B-mediated cytotoxicity in a breast cancer model, and CXCR6-dependent tumor control occurs in hepatic metastasis [164,165,166]. ILC2s display tumor-type–specific functions across distinct cancer models. In lymphoma, they secrete IL-10 and promote IL-33 expression, which drives CXCR2 and its ligands, contributing to tumor cell apoptosis. In CRC, ILC2s enhance anti-tumor immunity by activating CD8^+^ T cells. In melanoma, they produce IL-5, which facilitates the recruitment of eosinophils into the TME and supports immune-mediated tumor control [147,149,152]. ILC3s exert TRAIL-dependent cytotoxicity, particularly in hepatocellular carcinoma and melanoma, and promote tumor suppression by releasing effector cytokines, including IFN-γ, GM-CSF, IL-8, IL-22, and TNF-α [162].

**Figure 3 cells-14-00812-f003:**
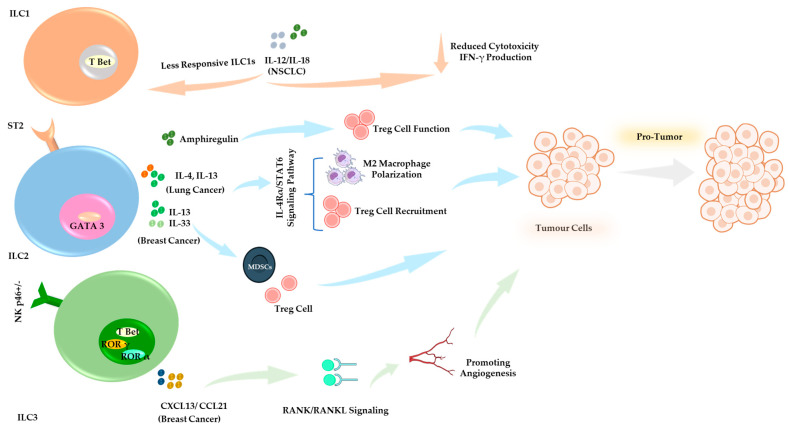
Pro-Tumor Functions of ILCs in the TME. ILCs contribute to tumor progression and immune evasion. ILC1s in NSCLC reduce cytotoxicity and impair IFN-γ production. IL-12/IL-18 signaling contributes to diminished responsiveness and effector function [127]. ILC2s secrete IL-13, IL-4, and amphiregulin, contributing to tumor-associated immune suppression [134,135,136,140,167]. In breast cancer models, IL-33 and IL-13 signaling promote the expansion of Tregs and MDSCs, thereby facilitating tumor progression [146]. ILC3 in breast cancer secretes CXCL13 and CCL21, promoting stromal interactions and tumor-associated inflammation, contributing to tumor progression and metastasis [151,160,168,169,170].

## 5. Mechanisms of Endogenous DC-ILC Crosstalk: Biological and Therapeutic Implications

Interaction between DCs, the professional antigen-presenting cells, and ILCs, a group of cells lacking adaptive antigen receptors that function similarly to T-lymphocytes in innate immunity [8,171] is bidirectional and does not only rely on direct contact but can also occur via soluble factors. The activation of ILCs and the maturation of DCs occur through cross-talk between these two cell types. These interactions influence the quality and intensity of subsequent immune responses to tumor cells and infectious agents [3].

NK cells are unique in killing target cells without prior sensitization. They are important for controlling certain types of viral infections and tumor surveillance [104]. NK cells are essential components of the innate immune system that function similarly to CD8+ T cells in eliminating abnormal cells, such as those infected by viruses or transformed into cancer cells. Unlike CD8+ T cells, NK cells recognize changes in self-MHC class I expression and stress signals in damaged cells. They are equipped with various activating and inhibitory receptors to carefully regulate their activity, making them a valuable tool in cancer immunotherapy. Although NK cells are a major focus in cancer immunotherapy, this study will focus more on the functions and potential of another subset of ILCs, specifically ILC1, ILC2, and ILC3, contributing to immune responses [117].

Mo-DCs have been shown as the most potent inducers of NK cell proliferation and cytotoxic activity, the functions that may extend to ILC1s due to their shared IFN-γ-driven responses [3]. Human dermal DCs demonstrate intermediate activation potential, whereas LCs require exogenous IL-2 and IL-12 to stimulate NK cells effectively. This suggests a similar cytokine-dependent activation pattern for cytotoxic ILC3s, including NCR^+^ ILC3s, which are dependent on IL-2 and IL-12 for optimal activation and cytotoxic function [3,172,173].

Previous studies highlighted this relation in the initiation of anti-tumor responses. After the activation of DCs following tumor development, they activate ILC1 by secreting IL-12. Upon activation, ILC1s generate and secrete IFN-γ and TNF-α, which can subsequently activate different groups of cells with anti-tumor activity, such as NK cells [174]. In addition to the secretion of IL-12, the expression of IL-15Rα seems necessary for NK cell activation. Other cytokines, such as IL-2, IL-18, and type I IFN, are vital in this cross-talk [3]. This cross-talk exhibits anti-tumor effects and initiates immune responses against infections. DCs not only enhance the cytotoxic potential of NK cells but also rely on them for immune surveillance. In response, NK cells contribute to immune homeostasis by identifying and eliminating infected DCs with impaired antigen-presenting function, thereby preventing tumor and viral immune evasion [175]. Recognition of DCs by NK cells can occur via a receptor named NKp30 (classified in the natural cytotoxicity receptor family). This connection activates NK cells, which subsequently engage in DC ‘editing’, selecting DCs with the most efficient antigen-presenting capability [176]. This editing process, mediated by NK cells, can eliminate DCs with impaired antigen processing and T-cell priming abilities, thereby selecting the most functionally competent DCs [177]. These findings underscore the significance of this bidirectional cross-talk in the comprehensive activation of various immune agents against tumors. So, any therapeutics and vaccines with the potential for activating and improving this cross-talk can be considered as an option, like using exogenous ILs that are involved in this cross-talk (such as IL-2 and IL-18) or applying nucleotide-based platforms (such as mRNA-based ones) to increase the expression of these cytokines or even overexpression of receptors which enhance cell recognition process between engaged cells.

According to a study by Campana et al., there may be a more efficient interaction between ILC3s and DCs, which could serve as an early and potent activator of DCs. DCs have the capacity to stimulate ILC3s through signaling pathways that bind to DNAX accessory molecule 1 (DNAM-1), also known as CD226 [176]. DNAM-1/CD226 belongs to the immunoglobulin superfamily and is predominantly expressed in NK cells, CD8^+^ T cells, CD4^+^ T cells, monocytes, and platelets in humans and mice. This molecule plays a critical role in the cytotoxic function of these immune cells, contributing to immune surveillance and tumor elimination [178]. IL-17 and IL-22 secretion by ILC3 is essential in regulating DC function. For instance, IL-17 contributes to the upregulation of co-stimulatory molecules on DCs, thereby enhancing their capacity to stimulate T-cell activation [176,179]. Additionally, DCs, upon sensing microbial signals, produce IL-23 and IL-1β, which activate ILC3s within the TME. Activated ILC3s secrete CXCL10, facilitating the recruitment of CD4^+^ and CD8^+^ T cells via CXCR3 signaling. These effector T cells, along with NK cells, exert cytotoxic effects by releasing IFN-γ, granzyme B, and perforin, thereby contributing to tumor suppression [180].

The cross-talk between ILCs and DCs can also affect the nature of ILCs. This dynamic crosstalk influences immune responses by modulating antigen presentation, cytokine production, and T-cell differentiation, ultimately impacting inflammation and tissue homeostasis [181]. ILC2s are known for their role as antiparasitic defense and their relationship with allergic reactions and tissue repair. ILC2-derived IL-13 has been shown to trigger the migration of DCs into the local draining lymph node, where these DCs guide naive T cells to develop into Th2 cells [182]. In Th2-mediated immunity, Th2 cells predominantly produce cytokines such as IL-4, IL-5, IL-9, and IL-13, contributing to eosinophil recruitment, IgE class switching, and mucus production [183,184]. In addition, ILC2-derived IL-13 promotes the release of the chemokine CCL17 from DCs, attracting CCR4^+^ memory Th2 cells to sites of allergen exposure. It has been suggested that IL-13 influenced lung DC migration partly by modifying the PGE_2_-EP4 pathway [185]. However, the precise mechanisms IL-13 regulates DC migration remain incompletely understood [186].

DCs and macrophages generate the TNF family member TL1A when TLR and Fc receptors are cross-linked. TL1A is important for modulating adaptive immune responses because it co-stimulates T cells. Studies indicate that TL1A synergizes with IL-25 in vivo, enhancing the expansion, survival, and functionality of ILC2s. Furthermore, TL1A’s role in activating ILC2s sheds light on the interaction between ILC2s and activated myeloid cells [187].

Recent studies highlighted the critical function of type I and II IFNs and IL-27 in negatively regulating ILC2s to limit type 2 immunity and its associated conditions. pDCs have been reported to play a crucial role in suppressing the function and survival of ILC2s during allergic pulmonary inflammation. Activated pDCs produce IFN-α, inhibiting ILC2 proliferation and increasing their apoptosis rate. Similarly, NK cells activated by polyinosinic–polycytidylic acid (pI:C) suppress the proliferation and cytokine production of ILC2s through IFN-γ during the early stages of lung inflammation, which recalls the classic antagonism between Th1 and Th2 pathways. Given that interferons are produced by various cell types, including Th1 cells and natural killer T (NKT) cells, it is reasonable to infer that diverse cellular sources of these mediators could suppress ILC2s [174,188]. ILC2s may change to an ILC1-like phenotype in response to cytokines IL-1, IL-12, and IL-27 secreted by macrophages and DCs during chronic inflammatory conditions in the airways [181,189]. A subset of DCs, cDC1s, which appear to be a critical group of DCs in tumor defense, are abundant in the TME. IL-18 and IFN-α-activated NK cells attract iDCs and enhance the function of mature DCs, stimulating them to secrete CXCL9, CXCL10, and CCL5. This promotes the recruitment of effector immune cells, strengthening the anti-tumor immune response [190].

After NK cells mature DCs, they release cytokines such as IL-2, IL-12, or IL-18, stimulating NK cells to produce other factors, including IFNγ, TNF-α, and GM-CSF. These factors can also help in maturing DCs. In addition to this type of connection, NK cells can also mature DCs via the ligation of CD40/CD40L. This ligation can help upregulate membrane-bound IL-15 expression on DCs, promoting NK cell proliferation [190].

Integrating ILC modulation into DC-based vaccine strategies represents an exciting advancement in cancer immunotherapy. These strategies utilize the innate immune system’s ability to influence adaptive responses, address existing challenges, and provide more effective anti-tumor immunity. Further investigation into the mechanisms that regulate ILC-DC interactions and their effects on the TME will be crucial for fully realizing the therapeutic potential of this approach.

## 6. Modulatory Effects of DC Vaccines on ILC Functionality Within Tumor Microenvironments

DC vaccines represent an advanced strategy in cancer immunotherapy due to their ability to induce potent anti-tumor immune responses. By priming DCs with tumor antigens, these vaccines enhance immune recognition and facilitate the elimination of cancer cells [191]. Immunotherapies like DC-based vaccines can modify ILCs by changing their traits and functions, potentially guiding them to enhance anti-tumor immunity instead of fostering cancer. This strategy may serve as a significant tool for cancer treatment and research, providing a more customized immune response.

NK cells are essential for the success of DC-based cancer vaccines. In the absence of NK cells, the ability of the vaccine to fight tumors is weakened. However, there is still limited research on how DC vaccines interact with other ILCs [192]. A critical aspect of the DC vaccine involves the targeted engagement of distinct DC subpopulations to enhance their unique immunomodulatory properties. Among the various DC subsets, cDC1s have emerged as key players in shaping anti-tumor immunity [193]. cDC1-derived cytokines, particularly IL-12 and type I IFNs, are pivotal in promoting additional immune effector populations, including Th1 cells, NK cells, NKT cells, and ILC1, reinforcing a comprehensive anti-tumor response [194].

ILC2s are known to express molecules associated with T cell suppression, and their increased frequency in the circulation of gastric cancer patients is correlated with elevated levels of IL-33, IL-25, IL-5, and IL-4, alongside a paradoxical decrease in IL-13. Notably, ILC2s can modulate MDSCs and M2 macrophages by upregulating inducible nitric oxide synthase (iNOS) and arginase-1 (Arg1). Since Arg1 depletes L-arginine, a key amino acid required for T cell function, its presence in the TME leads to defective T cell receptor expression and cell cycle inhibition. This mechanism is particularly relevant in the DC vaccination, as the success of this immunotherapy depends on functional and proliferating T cells. The presence of Arg1-expressing ILC2s in the TME may contribute to a diminished T cell response, limiting the efficacy of DC-based vaccines [107].

In a study conducted by our team, we investigated the effect of DC vaccination on the numbers of ILC2 and their cytokine secretion. We found that the number of ILC2s increased in the draining lymph nodes, spleen, and lungs after vaccination, particularly in a pulmonary metastatic murine melanoma model. This suggests that the vaccine might interact with ILC2s, influencing the movement or proliferation of these cells and possibly enhancing the immune response. Moreover, we observed a type of communication between DCs and ILC2s, particularly involving the cytokines IL-5 and IL-13, which indicates an active immune response triggered by the vaccine [4,10,63].

We also demonstrated communication between DC-based vaccines and ILC3 subpopulations in tumor-free and tumor-bearing mice [11]. In the local draining lymph node of naïve mice (mice without cancer), there were more ILC3s (NCR^+^ and NCR^−^ subsets), while the total number of ILC3s in the spleen did not change significantly post-vaccination. However, there was an increase in the production of IL-17 and IL-22 by the ILC3s in the spleen of vaccinated mice.

We further examined how immunization with DCs affected ILC3 responses in the tumor-bearing animals [11]. Mice were vaccinated with DCs and challenged with B16F10 melanoma cells. We demonstrated a shift in ILC3 subpopulations in the lungs, with a decrease in NCR^−^ ILC3s and an increase in NCR^+^ ILC3s, with no changes in cytokines produced by ILC3s in the lungs. This could be attributed to the switching of NCR^−^ILC3s to NCR^+^ ILC3s. During this transition, ILC3s might not have been able to produce cytokines immediately, which could explain the lack of observed cytokine changes in the lungs [11]. NCR^+^ ILC3s express the T-bet transcription factor, which drives their ability to promote Th1-like immune responses. Also, NCR^+^ ILC3s are capable of producing CXCL10, a chemokine that facilitates the recruitment of CD8^+^ T cells into the TME, thereby enhancing anti-tumor immunity [11,195].

The plasticity of ILCs and their ability to switch between functional states in response to microenvironmental signals make them an attractive target for DC-based vaccine strategies [11] (Figure 4). DC-based vaccines can be engineered to secrete specific cytokines (e.g., IL-12, IFN-γ) that shift ILC function toward an anti-tumor phenotype, promoting ILC1-like activity to enhance cytotoxic T-cell responses [196].

Collectively, these insights highlight the critical role of DC–ILCs interactions in shaping immune responses, from maintaining homeostasis to influencing the efficacy of immunotherapeutic interventions (Figure 4).

## 7. Strategic Enhancements of DC Vaccines: Targeting ILCs for Optimized Anti-Tumor Immunity

As key antigen-presenting cells within the immune system, DCs can initiate, direct, and modulate innate and adaptive immune responses. This unique ability positions DCs as an optimal candidate for developing novel therapeutic strategies in immunotherapy. By utilizing their antigen-presenting capabilities and role in immune education, DC-based vaccines have emerged as a compelling preventive strategy against cancer [10]. To enhance the efficacy of this type of vaccine, understanding the interactions and cross-talk between DCs and ILCs is critical due to their bidirectional nature. Improving the efficacy of DC-based vaccines can be indirectly achieved by enhancing ILC functions. Stimulation of DCs can cause them to secrete IL-12, which activates NK cells, cytotoxic T lymphocytes, and Th1 cells, leading to the production of IFN-γ [197,198]. Furthermore, IL-12 can directly affect tumor cells, forcing them to increase the expression of antigen MHC-1 on the cell surface, which can facilitate tumor identification [199]. In addition to IL-12, IL-18, which is expressed by different immune cells, including DCs, also has anti-tumor activity [200]. Engineering DCs to produce IL-12, either alone or in combination with IL-18, represents a novel approach to enhancing the efficacy of DC-based vaccines, as investigated by Mierzejewska et al. According to their study, the application of DCs engineered to produce IL-12 alone or in combination with IL-18 can enhance the filtration and activity of CD4+ and CD8+ T lymphocytes in the TME and tumor-draining lymph nodes [199]. Based on these findings, IL-12 plays a crucial role in mediating anti-tumor responses through multiple pathways, including the activation of immune effector cells, the enhancement of cytotoxic activities, and the direct inhibition of tumor cell proliferation. One potential strategy to enhance the efficacy of DC vaccines is the genetic engineering of DCs to secrete elevated levels of IL-12, a cytokine known to promote Th1 polarization, enhance CTL responses, and activate NK cells. This approach may be further potentiated when combined with other immunotherapeutic modalities, such as immune checkpoint blockade (e.g., anti-PD-1, anti-CTLA-4) to alleviate T cell exhaustion and restore effector function; adoptive cell therapies, including CAR-T or TCR-engineered T cells, which benefit from a pro-inflammatory, IL-12-rich TME; and TLR agonists, which can serve as adjuvants to amplify DC maturation and cytokine production. Additionally, co-administration with oncolytic viruses may enhance local inflammation and antigen release, further synergizing with IL-12-driven immune activation. Such combinatorial strategies aim to overcome immunosuppressive barriers within the TME and augment the therapeutic potency of DC-based cancer vaccines. Furthermore, IL-18 can prime NK cells, which allows these cells to recruit DCs using CCL3 and CCL4, subsequently inducing cDC1s to produce CXCL9, CXCL10, and CCL5. The produced chemokines present an anti-tumor effect by attracting expanded cytotoxic T cells to the tumor site [201]. The cross-talk between DCs and NK cells is not exclusively dependent on chemokines; DCs can also modulate NK cells through adhesion molecules such as CD155 and CD112 [201]. Exploring strategies to regulate the expression of these molecules may provide new approaches to enhance the efficacy of DC vaccines. Adding different protein-encoding genes to increase the activity of DCs has been previously observed in other studies. Yan et al. developed a nanovaccine incorporating the CD155 gene within a liposome. Their findings demonstrated that this nanovaccine promotes DC maturation and differentiation through a synergistic pathway involving TLR4 and macrophage galactose-type lectin (MGL), facilitating antigen recognition and immune activation [202]. As we know, this molecule also plays a role in cross-talk between DCs and NK cells. Applying this method with DC vaccines is also important and needs more studies. Employing mRNA technology to stimulate and increase the efficacy of DCs is a new and promising approach for developing immunotherapy methods, helping enhance antigen presentation, and their effectiveness has also been evaluated previously [203]. In a previous study, Pfeiffer et al. found that human MO-DCs, which were matured with a cytokine cocktail and transfected via mRNA electroporation with constitutively active IκB kinase (caIKK) mutants, displayed elevated levels of maturation markers. These cells also produced more significant amounts of cytokines, including IL-12p70, and exhibited an improved ability to activate and expand cytotoxic T lymphocytes with a memory-like phenotype [203,204]. Since IL-12 can help activate NK cells, utilizing this type of mRNA may assist in indirectly activating NK cells. This theory was the focus of Bosch et al.’s research. They aimed to demonstrate whether the transfection of DCs with mRNA encoding IKKβ can activate autologous NK cells. They observed this phenomenon through the upregulation of CD54, CD69, and CD25 on the NK cells, as well as their ability to secrete IFN-γ and their heightened cytolytic activity [203]. These studies suggest that mRNA can manipulate and engineer immune gene expression, influencing cross-talk between the immune cells and enhancing their activation and maturation. Utilizing this nucleotide-based platform enables cells to overexpress specific cytokines, such as IL-12 or IL-18, along with other molecular markers facilitating intercellular communication.

IL-12, IL-15, and IL-18 predominantly drive ILC1 activation. mRNA-engineered DC vaccines can be designed to overexpress these cytokines, potentially augmenting ILC1 expansion, survival, and cytotoxic activity [205].

Combining DC vaccines with agents like IL-33 or other ILCs stimulators may synergistically boost the immune response against tumors [206]. For example, IL-33 is a strong activator of ILC2s and ILC3s, stimulating ILC2s to secrete type 2 cytokines and prompting ILC3s to produce pro-inflammatory cytokines. In the TME, recruitment and activation of ILCs by IL-33 can enhance anti-tumor immunity or facilitate tumor progression, depending on the context and balance of the immune response [207,208].

Engineering DCs to express co-stimulatory molecules, such as CD80 and CD86, can enhance their interactions with ILCs, particularly ILC2s and ILC3s, thereby promoting more effective T cell activation [31]. CD80 and CD86 expression on DCs can facilitate ILC activation and function by providing analogous co-stimulatory signals. This interaction may stimulate ILCs to secrete cytokines and effector molecules, contributing to the immune response [209].

Modifying DC vaccines to produce specific chemokines that recruit ILCs into the TME can enhance localized immune responses. CCL20 can attract ILC3s, potentially contributing to maintaining a pro-inflammatory environment [210]. Chemokines, particularly CCL20, are crucial in recruiting ILC3s, which secrete key cytokines such as IL-22 and IL-17, shaping the immune response within the TME [211]. Additionally, ILC3s produce factors that can directly inhibit tumor growth or induce apoptosis in cancer cells, such as Granzyme B and IFN-γ [113,180]. These mediators function through cytotoxic activity, immune cell recruitment, and modulation of the TME, collectively contributing to tumor suppression and immune surveillance.

These strategies utilize the innate immune system’s ability to influence adaptive responses, enhancing anti-tumor immunity (Figure 5). Ongoing research into the regulatory mechanisms that govern ILC-DC interactions and their effects on the TME will be essential for optimizing this approach’s therapeutic potential.

## 8. Future Perspectives and Conclusions: Advancing the Frontier of DC Vaccines and ILC Research in Cancer Immunotherapy

Clinical trials have demonstrated DC vaccine safety and tolerability across diverse tumor types, yet their efficacy remains limited, mainly due to the suboptimal immune activation and immunosuppressive TME. This review focused on how DC vaccination influences NK cells, ILC2s, and ILC3s and their cytokine production, highlighting the complex relationship between DC-based vaccines and ILCs.

While traditional DC vaccine strategies have primarily aimed to activate T cell responses, recent platforms provide new opportunities to reprogram DCs to enhance the modulation of TME and the regulation of innate immunity. Optimizing DC vaccines to promote ILC recruitment and activation could improve therapeutic efficacy [212]. However, several challenges must be addressed, including ILC plasticity, TME heterogeneity, and the need to overcome immunosuppressive signaling pathways [97,98,108].

A comprehensive strategy must integrate enhanced ILC activation, improved ILC recruitment, and TME reprogramming to preserve effective anti-tumor immunity. Reprogramming DCs to reestablish their antigen-presenting function and neutralize immunosuppressive signals is crucial for optimizing vaccine efficacy. Engineering DC vaccines to overexpress cytokines may reinforce the potential to enhance ILC-mediated anti-tumor responses significantly. Engineering DCs to secrete chemokines such as CXCL9, CXCL10, and CCL5 could facilitate the recruitment of ILCs. The TME often exerts strong immunosuppressive pressures by accumulating Treg cells, MDSCs, and inhibitory cytokines. These elements can impair ILC function and limit their cytotoxic potential. Therefore, successful implementation of DC-ILC-based immunotherapy requires the development of combinatorial strategies that not only attract effector cells but also reprogram the immunosuppressive environment of the TME, such as incorporating checkpoint blockade, cytokine modulation, or TME-targeted metabolic interventions. Overcoming TME-induced immunosuppression is crucial for the success of DC-ILC-based therapies [201]. TGF-β signaling [118] is a major barrier that suppresses ILC1 cytotoxicity and weakens anti-tumor immunity. Targeting TGF-β inhibition may help preserve ILC1 function, allowing for more effective immune responses. Beyond molecular modifications, advanced immune profiling techniques, such as single-cell RNA sequencing, will provide deeper insights into the regulatory pathways controlling DC–ILC interactions, paving the way for more precise and customized immunotherapeutic strategies. Given the immunosuppressive nature of the TME, checkpoint inhibition approaches, such as PD-1 blockade and LAG3/TIM-3 inhibitors, present a compelling strategy with which to restore ILC function by counteracting tumor-induced immune suppression and enhancing anti-tumor immunity [117]. These inhibitory receptors are often upregulated during chronic inflammation and within the tumor microenvironment, impairing the effector functions of ILC subsets. Targeting these suppressive pathways with checkpoint inhibitors can restore cytokine secretion, cytotoxic function, and the survival of ILCs. When used in conjunction with DC-based vaccines, these strategies may synergistically enhance immune activation by stimulating both innate and adaptive immune responses, thereby improving tumor elimination in immunologically non-inflamed or poorly infiltrated tumor settings.

The future of DC-ILC immunotherapy will depend on comprehensive preclinical and clinical validation to determine the most effective strategies for integrating ILC-targeting approaches into DC vaccine platforms. Future clinical trials should prioritize monitoring ILC dynamics post-vaccination to ensure optimal engagement of innate and adaptive immunity.

Advancing DC-ILC-based immunotherapies could significantly enhance anti-tumor responses and immune activation, paving the way for next-generation treatments. By refining DC vaccine engineering and incorporating ILC-targeted strategies, next-generation immunotherapies may offer more precise, effective therapies. Advancing research into DC–ILC interactions will be crucial for shaping the future of personalized cancer immunotherapy and unlocking novel, more effective therapeutic strategies.

## Figures and Tables

**Figure 1 cells-14-00812-f001:**
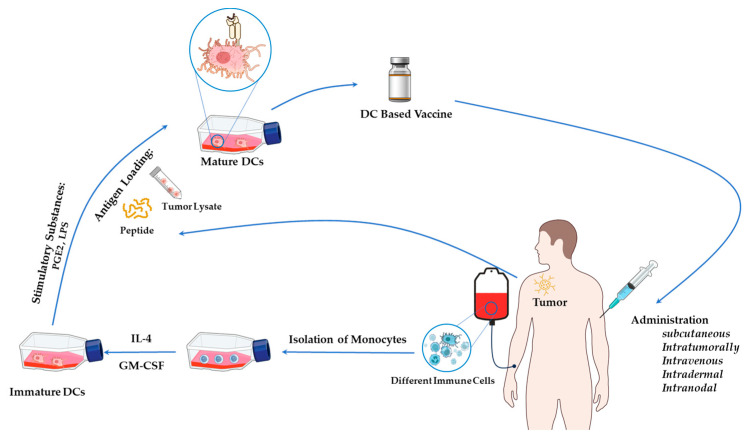
The standard procedure for generating DC vaccines.

**Figure 4 cells-14-00812-f004:**
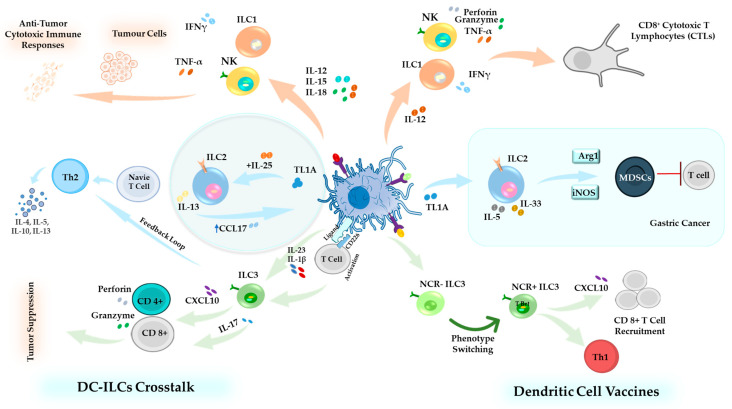
DC–ILCs Crosstalk in Homeostasis and Cancer Immunotherapy. Illustration of the multifaceted interactions between DCs and ILCs. Upon activation, DCs release cytokines such as IL-12, IL-15, and IL-18, which stimulate ILC1s and NK cells to produce IFN-γ and TNF-α, thereby promoting cytotoxic and anti-tumor responses. ILC2s, influenced by IL-13 and TL1A, support Th2-mediated immunity [187]. ILC3s, activated via CD226 (DNAM-1) signaling, secrete IL-17, enhancing T cell activation. Additionally, ILC3s activated by IL-23 and IL-1β produce CXCL10, which stimulates CD8^+^ and CD4^+^ T cells, contributing to tumor suppression [178,195]. DC-based cancer vaccines activate cDC1s, which release IL-12 and type I IFNs, driving ILC1 differentiation and enhancing cytotoxic T-cell responses. NK cells further enhance vaccine efficacy by amplifying anti-tumor immunity [194]. ILC2s and MDSCs promote T cell suppression, potentially reducing vaccine effectiveness [107]. ILC3s, stimulated by DC-derived signals, are reprogrammed toward a more anti-tumor NCR^+^ phenotype. These NCR^+^ ILC3s express T-bet, promoting Th1-like immune responses [11,195].

**Figure 5 cells-14-00812-f005:**
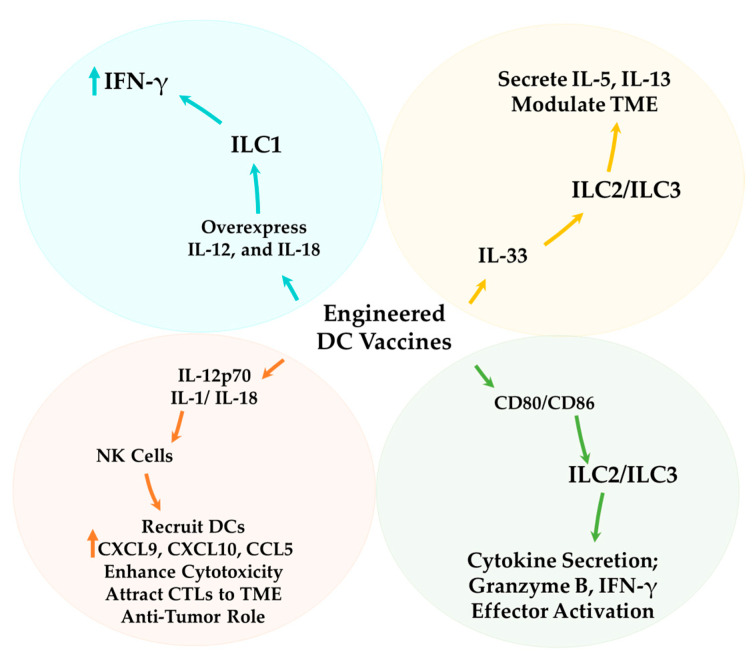
Strategic Enhancements of DC Vaccines through Targeting ILCs. DC-based cancer vaccines can be optimized by utilizing ILCs interactions. IL-12 and IL-18, secreted by engineered DCs, stimulate NK cells, which enhance cytotoxicity, support CTL expansion within the TME, and recruit additional DCs [200]. Engineered DC vaccines further enhance ILC1 expansion within the TME. Additionally, IL-33 activation of ILC2/ILC3 modulates the TME. DCs engineered to express co-stimulatory molecules (CD80/CD86) strengthen interactions with ILC2s and ILC3s, promoting cytokine secretion and effector activation [209]. Modified DC vaccines facilitate the recruitment of ILC3s, which secrete IL-22 and IL-17, contributing to a pro-inflammatory response in the TME [211]. ILC3s, in turn, release Granzyme B and IFN-γ, enhancing tumor suppression through cytotoxic activity, immune cell recruitment, and apoptosis induction [113,180].

## Data Availability

Not applicable.

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
