# Peer review of "Dendritic Cell-Based Cancer Vaccines: The Impact of Modulating Innate Lymphoid Cells on Anti-Tumor Efficacy"

_cells, 2025, doi:10.3390/cells14110812_

Round 1
Reviewer 1 Report
Comments and Suggestions for Authors
The submitted review is extremely interesting and detailed. The authors put the attention of different aspects of the cross-talking among the different cell subtypes, and went deep inside the different aspects of the impact of ILC subpopulations on the effects of DC-based vaccines and viceversa.
Minor revision:
lines 239-242: spaces among lines should be reduced.
line 737: INF should be IFN
Reviewer 2 Report
Comments and Suggestions for Authors
This manuscript presents a comprehensive and detailed review of the interplay between DC vaccines and ILCs in cancer immunotherapy. It is well-referenced, timely, and provides a strong overview of both the biological foundations and clinical implications of DC-ILC interactions. The authors offer valuable perspectives on enhancing DC vaccine efficacy through the modulation of ILCs.
However, the manuscript would benefit from further focus and condensation. Specifically, Sections 2 and 4 could be more concise, emphasizing only the aspects directly relevant to DC-ILC interactions. Additionally, figures should be more clearly referenced within the text, with their key messages explicitly explained at the point of citation to enhance clarity and integration.
Reviewer 3 Report
Comments and Suggestions for Authors
In the review “Dendritic Cell-Based Cancer Vaccines: The Impact of Modulating Innate Lymphoid Cells on Anti-Tumor Efficacy” by Yeganeh Mehrani et al., the authors discuss several aspects of the improvement of DC-based cancer vaccines via the modulation of innate lymphoid cells. The chapters of the reviews are “Characterization of Dendritic Cell Vaccines: Production Techniques and Efficacy Against Neoplasms”, “Limited Effectiveness of DC Vaccines and Challenges Faced in Clinical Trials”, “ILCs in Oncology”, Mechanisms of Endogenous DC-ILC Crosstalk: Biological and Therapeutic Implications”, “Modulatory Effects of DC Vaccines on ILC Functionality within Tumor Microenvironments”, and “Strategic Enhancements of DC Vaccines: Targeting ILCs for Optimized Anti-Tumor Immunity”. The review has five figures and 274 references.
The manuscript seems well organized and written, however, there are several aspects the authors should improve and/or respond to. The review could be improved by shortening.
- Chapter 2 is a long one with DC-related references of the time between 1997 and 2012. The chapter should be shortened and really focused according the title “Characterization of Dendritic cell vaccines: Production Techniques and Efficacy Against Neoplasms”.
- Chapter 2 starts with a section of text without subheadings (line 63-157); then subheadings start in line 158-242, in line 243 starts again a part without subheading. The authors should try to use subheadings for all the text and in all chapters. This makes it easier to follow the red thread of the review.
- “Efficacy against neoplasms” is part of the title of chapter 2, “limited effectiveness of DC vaccines” is part of the title of chapter 3. The authors should avoid repetitions in the chapter headings and make the common thread clearer.
- Chapter 3 includes DC/NK cells crosstalks, but the title of this chapter was “Limited Effectiveness of DC Vaccines and Challenges Faced in Clinical Trials”. Should NK cell be better part of chapter 5 (Mechanisms of Endogenous DC-ILC Crosstalk: Biological and Therapeutic Implications)?
- Line 82 and Table 1 and line 126-134: Different types of DC are listed by the authors together with follicular dendritic cells. Since follicular dendritic cells are not derived from the bone-marrow hematopoietic stem cell, but are of mesenchymal origin, they should not be designated as antigen-presenting DC. Unlike other DCs, FDCs even lack MHC class II antigen molecules. Why is it necessary to mention this type of stromal cell in the context of the review? If FDC are necessary to mention, why other Germinal-center-associated stromal cells (PMID: 38431843) are lacking in the review?
- Line 66-68: What is meant by the statement “Immature DCs express low toll-like receptors (TLRs) and MHC I and II molecules, limiting their antigen-presenting capacity as they migrate beyond lymphoid organs.”?
- Line 607-609: I do not understand the connection how ILC2 “benefit certain cancers by inducing tumor cell apoptosis and limiting tumor growth”
- Line 238: In the part about “Allogeneic DC vaccines” the authors not even mention HLA haplotypes.
- The paper has 274 references. However, not all of these references seems necessary. Please revise your references! As an example: line 722 “Interaction between DCs, the professional antigen-presenting cells (229,230),…” Is it really necessary to give references for DC as APC?
- Line 48: Ref. (5) is included for the statement “Recent studies emphasize the critical role of innate lymphoid cells … in shaping immune responses within the tumor microenvironment”. However, ref. (5) does not mention innate lymphoid cells.
- Line 164: Ref. 58 from 1997 is given for the procedure of DC maturation, however, DC are not part of this paper.
- The citation style has to be revised, e.g.
journal is lacking in Ref. 42,
Volume and pages lacking in Ref.46,
64: journal, year?
Ref. 55: Bjö NR, Clausen E, Stoitzner P, Romani N, Clausen BE- no result in PubMed
many references are “available from other sources…”
ref. 70: “MOLECULAR BIOTECHNOLOGY. 2001”?
- The abbreviations should be arranged in an alphabetical order.
